# Do statistical heterogeneity methods impact the results of meta- analyses? *A meta epidemiological study*

**Samer Mheissen**[1]*, **Haris Khan**[2], **David Normando**[3], **Nikhillesh Vaiid**[4], **Carlos Flores-Mir**[5]

**1** Private Practice, Damascus, Syria, **2** Department of orthodontics, CMH institute of dentistry Lahore, National University of Medical Sciences, Punjab, Pakistan, **3** Department of Orthodontics, Federal University of Pará, Belém, Brazil, **4** Visiting Professor, Faculty of Dentistry, National University of Singapore, Queenstown, Singapore, **5** Division of orthodontics, School of Dentistry, University of Alberta, Edmonton, Canada

* Mheissen@yahoo.com

**Data Availability Statement:** All relevant data are within the paper and its Supporting Information files.

## Abstract

### Background

Orthodontic systematic reviews (SRs) use different methods to pool the individual studies in a meta-analysis when indicated. However, the number of studies included in orthodontic meta-analyses is relatively small. This study aimed to evaluate the direction of estimate changes of orthodontic meta-analyses (MAs) using different between-study variance methods considering the level of heterogeneity when few trials were pooled.

### Methods

Search and study selection: Systematic reviews (SRs) published over the last three years, from the 1st of January 2020 to the 31st of December 2022, in six main orthodontic journals with at least one MA pooling five or lesser primary studies were identified. Data collection and analysis: Data were extracted from each eligible MA, which was replicated in a random effect model using DerSimonian and Laird (DL), Paule–Mandel (PM), Restricted maximum-likelihood (REML), Hartung Knapp and Sidik Jonkman (HKSJ) methods. The results were reported using median and interquartile range (IQR) for continuous data and frequencies for categorical data and analyzed using non-parametric tests. The Boruta algorithm was used to assess the significant predictors for the significant change in the confidence interval between the different methods compared to the DL method, which was only feasible using the HKSJ method.

### Results

146 MAs were included, most applying the random effect model (n = 111; 76%) and pooling continuous data using mean difference (n = 121; 83%). The median number of studies was three (range 2, 4), and the overall statistical heterogeneity ($I^2$ ranged from 0 to 99% with a median of 68%). Close to 60% of the significant findings became non-significant when HKSJ

**Funding:** The author(s) received no specific funding for this work.

**Competing interests:** The authors have declared that no competing interests exist.

was applied compared to the DL method and when the heterogeneity was present $I^2 > 0\%$. On the other hand, 30.43% of the non-significant meta-analyses using the DL method became significant when HKSJ was used when the heterogeneity was absent $I^2 = 0\%$.

## Conclusion

Orthodontic MAs with few studies can produce different results based on the between-study variance method and the statistical heterogeneity level. Compared to DL, HKSJ method is overconservative when $I^2$ is greater than 0% and may result in false positive findings when the heterogeneity is absent.

## Introduction

The publication of systematic reviews (SRs) in orthodontics has increased exponentially in recent years [1, 2]. A properly conducted SR requires many rigorous steps, starting with a comprehensive search for the available evidence, selecting the eligible studies, extracting data, and finally synthesizing the data [3, 4]. To guarantee valid results of an SR, authors should follow a proven rigor SR methodology that will maximize the proper consideration of the available evidence. Researchers generally use a meta-analysis (MA) to synthesize data from individual studies and provide the reader with summarized quantitative results. MA can facilitate interpreting the results by providing a cumulative effect measure. Selecting an appropriate statistical model is paramount for the validity of the MA.

Two main statistical models are applied in the MA: the fixed effect or the random effects model [5]. The fixed effect model assumes that all included studies in MA have the same true effect size, while the random effects model considers the effect size variation between the pooled studies in MA. In other words, the selection between the two models depends on the expected or measured effect size variation from study to study, which may result from differences (heterogeneity) in the participants' characteristics (age, gender, ethnicity, type of malocclusion, crowding level, etc.) and the implementation of the intervention (type, dose, activation of appliance, wearing time of appliance, follow up, etc.). The fixed effect model ignores the between-study heterogeneity or assumes it is nonexistent. In contrast, the random effects model quantifies the degree of statistical heterogeneity in MA and gives relatively larger weight to smaller studies than the fixed mode [6, 7]. Random effects model provides identical results to the fixed effect model if there is no heterogeneity. The random effects model implements different estimators for the between-study variance [3] to calculate confidence intervals in MA.

The simplest and most commonly applied between-study variance estimator is the DerSimonian and Laird (DL) approach [8, 9]. DL method is the default option in many popular statistical software packages such as Comprehensive Meta-analysis (CMA) [10] and Review Manager (RevMan) [11]. However, DL may lead to false positive results, particularly when the number of studies is small [12], the studies are unequal in size [13], and the heterogeneity is increased [14]. Many alternative methods exist to overcome the limitation of the DL method; the Paule–Mandel (PM) method [15], the Restricted maximum-likelihood (REML) method, the Hartung and Knapp [16] and the Sidik and Jonkman [17] (HKSJ) method. Many simulation studies [13, 18–20] investigated the effect of different between-study variance estimators on the pooled estimate, but the findings conflict. A previous study recommended the REML

and HKSJ over the DL methods [19], while PM was recommended in three simulation studies reported in a systematic review [21].

A previous study [22] revealed that approximately 65% of MAs in orthodontics have less than five trials. The potential impact of selecting a between-study variance estimator increases with smaller samples [23]. To address these shortcomings of pooling few trials in MAs in orthodontics a recent study investigated the use of the heterogeneity estimator corrected by Hartung Knapp on MAs in orthodontics [24]. Still, the authors included a wide range of pooled studies in the MA (from 3 to 45 per MA). This large variation in the primary studies can diffuse the true impact on smaller samples. Keeping this in mind, the present study aimed to investigate the impact of different between-study variance estimators, the degree of heterogeneity, and the sample size equality of pooled studies on the result of MAs in orthodontics with fewer trials (less than 5).

## Materials and methods

The reporting of this methodological study followed the reporting guidelines for methodological studies [25] which was adapted from Preferred Reporting Items for Systematic Reviews and Meta-Analyses (PRISMA).

### Eligibility criteria

SRs were included if they met the following criteria:

- Orthodontic SRs published between the 1st of January 2020 and the 31st of December 2022 in six journals: Progress in Orthodontics (PIO), European Journal of Orthodontics (EJO), Orthodontics & Craniofacial Research (OCR), American Journal of Orthodontic and Dentofacial Orthopedics (AJODO), The Angle Orthodontist (AO), and Korean Journal of orthodontic (KJO).

- These SRs should have included at least one meta-analysis with less than five primary studies reporting interventional procedures.

SRs of animal or laboratory trials, network meta-analyses, and MAs of incidence or prevalence were excluded. Scoping reviews, overviews, guidelines, and qualitative SRs were also excluded.

### Search and study selection

One author (xx) performed an electronic search in Medline via PubMed using MeSH terms and text word and journal websites to collect the relevant records (S1 Table). Two authors (xx, xxx) screened the titles and abstracts of the retrieved studies independently and in duplicate. Systematic reviews with meta-analyses related to orthodontics were included initially without consideration to the number of primary studies. Once the full version was obtained, the number of included primary studies per meta-analysis was evaluated. A discussion was completed until a consensus was reached in case of disagreement.

### Data collection process

The authors extracted the following characteristics of meta-analysis: the journal title, the effect measure (mean difference, Risk ratio, odds ratio, standardized mean difference), the MA model implemented (random, fixed), the number of included primary studies, the number of participants, the between-study variance estimator utilized (DL, REML, PM, HK, SJ), and the statistical software used.

For eligible meta-analysis; the number of each study group, mean, and standard deviation for continuous data, and number of events and non-events for dichotomous outcomes were extracted from the forest plots and entered in a Microsoft excel® (Microsoft, Redmond, Washington, USA) file for data presented in arm-level format. The mean, standard error or upper and lower confidence intervals were extracted if data were presented in contrast-level format.

## Statistical analysis and data synthesis

The characteristics of the included meta-analyses were summarized using median and inter-quartile ranges for continuous variables and frequencies for categorical variables. Each meta-analysis was replicated by one author (xx) using four different heterogeneity estimators (DL, REML, PM, HKSJ) available in Stata 15.1 software (Stata Corp, Texas, USA). The results of each replicated MA were compared to the original MA results using the same scenario to avoid the error in data extraction. Then from each MA output we collected; the overall effect size, the corresponding 95% confidence interval (CI), the P-value, and the level of statistical heterogeneity presented by $I^2$, Q, and $Tau^2$. Furthermore, the ratio between the confidence intervals of each estimator and the confidence interval of DL was calculated using the following equation:

$$CI^{Estimator} \ ratio \ = \frac{width \ of \ 95\% \ CI^{Estimator}}{width \ of \ 95\% \ CI^{DL}}$$

For the equality of trials' size, a percentage between the largest study and the smallest one in MA was calculated and if up to 20% the studies were considered to be equal to some extent, between 21–50% moderately unequal, between 51–100% unequal, and >100% substantially unequal.

The significance of the meta-analyses with the different heterogeneity estimators, P values, and confidence interval ratios for the different estimators were calculated and divided according to the statistical heterogeneity level expressed by $I^2$ statistics. The results were analyzed and visualized using the R statistical package (version 4.3.0) (R Foundation for Statistical Computing, Vienna, Austria). Finally, the feature selection algorithm Boruta was used to identify significant predictors associated with the significant change in the confidence interval from the DL method, and this was applicable to the HKSJ method.

## Results

### Meta-analyses characteristics

The selection process of the studies in the CONSORT format is illustrated in Fig 1. The excluded studies and the reasons for exclusion are clarified in the S2 Table.

A total of 146 meta-analyses with few studies were analyzed. The vast majority of the included MAs (n = 111; 76.03%) were conducted using the random effect model, and the output was expressed using the mean difference (n = 121; 82.88%). Likewise, most of the included MAs did not report the used heterogeneity estimator in their analysis (n = 99; 83.19%). Nevertheless, the DL method was reported in a few MAs (n = 14; 11.76%), and the other methods, such as HK, SJ, and REML, were barely reported. The median number of studies was 3 (IQR: 2 to 4), with a median number of participants 139 (IQR: 99, 249) per MA. The size of the studies per MA was equal in only 18% (21/146). The most frequently used software was RevMan (n = 112; 76.71%), followed by Stata and CMA software.

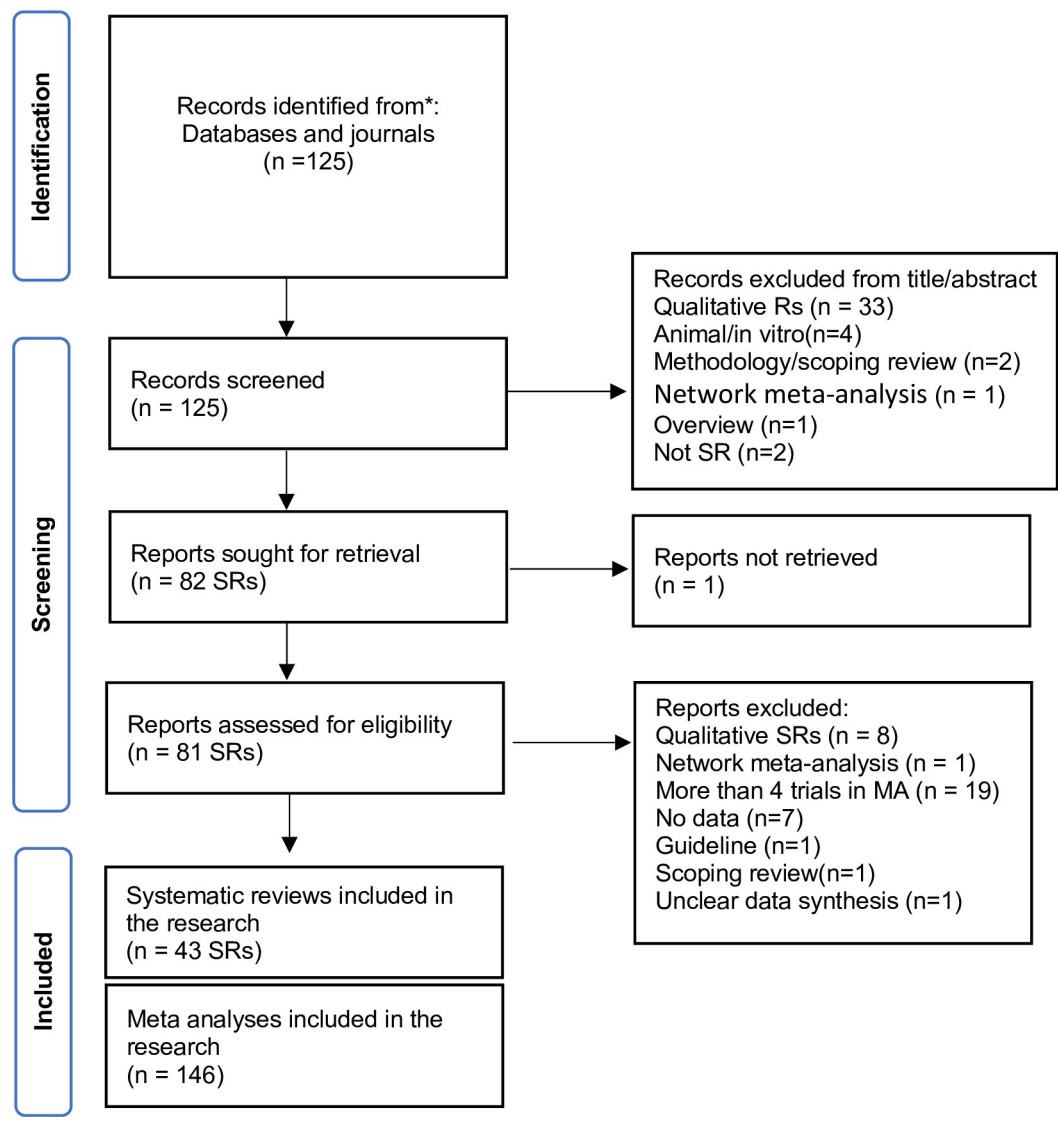

**Fig 1. The flowchart of the MAs selection process.**

The heterogeneity measures varied as for $I^2$ it ranged from 0 to 99% with a median of 68% (interquartile range (IQR): 0 to 78), for Q it ranged from 0 to 224.5 with a median of 2 (IQR: 0, 8), while for $t^2$(tau$^2$) it ranged from 0 to 75000 with a median of 0.0007 (IQR: 0 to 1) (Table 1).

### Replicating meta-analyses using the four heterogeneity estimator methods

The 146 MAs were replicated using the four heterogeneity estimators; DL, REML, PM, and HKSJ. (Table 2). The overall effect estimate did not change using the four mentioned estimators in MA. However, the change was obvious in the confidence interval's (CI) width and, subsequently, the findings' P-value and significance. REML and PM showed a slightly different confidence intervals and significance level. In contrast, HKSJ showed wider CI than DL, which was mostly double the width of DL confidence interval up to 6 times when $I^2$ was greater than 0. Also, HKSJ yielded wider confidence intervals when $I^2$ was 0, but this increase was smaller than MAs with higher $I^2$ levels. Interestingly, the confidence Interval using the HKSJ method

**Table 1. Characteristics of included meta-analyses.**

| | Characteristic | Overall number of meta-analyses, N = 146 |
|---|---|---|
| Journal, n (%) | AJODO | 33 (22.60%) |
| | Angle | 9 (6.16%) |
| | EJO | 51 (34.93%) |
| | KJO | 5 (3.42%) |
| | OCR | 36 (24.66%) |
| | PIO | 12 (8.22%) |
| Number of studies per meta, Median (IQR) | Median (IQR) | 3(2,4) |
| model, n (%) | fixed | 27 (18.49%) |
| | random | 111 (76.03%) |
| | NI | 8 (5.48%) |
| Q, Median (IQR) | | 2 (0, 8) |
| $I^2$, Median (IQR) | | 0.07 (0.00, 0.78) |
| $t^2$ DL, Median (IQR) | | 0 (0, 1) |
| participants number, Median (IQR) | | 139 (99, 249) |
| Equality of trials' size, n (%) | Equal | 21 (18%) |
| | Moderately unequal | 16 (14%) |
| | Unequal | 46 (39%) |
| | Substantially unequal | 34 (29%) |
| software, n (%) | Comprehensive Meta-Analysis | 12 (8.22%) |
| | R | 2 (1.37%) |
| | RevMan | 112 (76.71%) |
| | Stata | 19 (13.01%) |
| | NI | 1 (0.68%) |

was wider in all meta-analyses when $I^2$ was greater than 0, but it was narrower in (30/71; 42.25%) meta-analyses with a lack of heterogeneity ($I^2 = 0$) (Table 2 and Fig 2).

In regards to P-values and significance of the results, more than half (21/35; 60%) of significant findings of MAs with a statistical heterogeneity ($I^2>0$) using the DL method became non-significant using HKSJ. On the other hand, (7/47;30.43%) of non-significant results became significant, and (8/24;16%) of the significant results became non-significant when HKSJ was used instead of the DL method in meta-analyses with a lack of heterogeneity ($I^2 = 0$). In terms of other methods, there was a change in one significant result using the PM method (Table 3 and Fig 3).

The Boruta algorithm is a feature selection algorithm which measures the importance of each variable with respect to the outcome. The Boruta algorithm identified the heterogeneity measures (Q, $I^2$, and $t^2$) and the number of studies as significant predictors for the change in HKSJ confidence interval width compared to DL. In other words, the change in CI HKSJ/DL ratio was affected by the number of included studies in MA and the value of the statistical heterogeneity measures (Fig 4).

## Discussion

The DL method in random effect meta-analysis is a commonly used for calculating the confidence interval in orthodontic MAs. Although there was a lack of reporting of the between-study variance method in 83% (99/119) of random meta-analysis, 98% (97/99) of those MAs

**Table 2. Crosstabulation of different results of replicated meta-analyses using different methods of between-study variance with the level of statical heterogeneity expressed by $I^2$.**

| Characteristic | Overall number of meta-analyses, N = 146 | $I^2$ = 0, N = 71 | $I^2$ >0, N = 75 | p-value |
|---|---|---|---|---|
| **Effect size measure, n (%)** | | | | |
| MD | 121 (82. 88%) | 58 (81.69%) | 63 (84%) | |
| RR | 5 (3.42%) | 3 (4.23%) | 2 (2. 67%) | |
| SMD | 20 (13.70%) | 10 (14.08%) | 10 (13.33%) | |
| **Model, n (%)** | | | | 0.002[1] |
| fixed | 27 (18. 49%) | 20 (28. 17%) | 7 (9.33%) | |
| random | 111 (76.03%) | 50 (70.42%) | 61 (81.33%) | |
| NI | 8 (5.48%) | 1 (1.41%) | 7 (9.33%) | |
| **$tau^2$ Estimator, n (%)** | | | | >0.9[1] |
| DL | 14 (11.76%) | 5 (9.80%) | 9 (13.24%) | |
| HK | 1 (0.84%) | 0 (0%) | 1 (1.47%) | |
| MH | 2 (1.68%) | 1 (1.96%) | 1 (1.47%) | |
| REML | 1 (0.84%) | 1 (1.96%) | 0 (0%) | |
| SJ | 2 (1.68%) | 1 (1.96%) | 1 (1.47%) | |
| NI | 99(83.19%) | 43 (84.31%) | 56 (82.35%) | |
| **participants number, Median (IQR)** | 139 (99, 249) | 139 (89, 238) | 139 (105, 298) | 0.3[2] |
| $t^2$ DL, Median (IQR) | 0 (0, 1) | 0 (0, 0) | 0 (0, 4) | |
| **Q, statistics**, Median (IQR) | 2 (0, 8) | 0 (0, 1) | 8 (4, 17) | |
| **$I^2$ statistics**, Median (IQR) | | | 0.77 (0.45, 0.88) | |
| **RMEL/DL ratio**, Median (IQR) | 1.00 (1.00, 1.00) | 1.00 (1.00, 1.00) | 1.00 (0.99, 1.01) | 0.7[2] |
| **PM/DL ratio**, Median (IQR) | 1.00 (1.00, 1.00) | 1.00 (1.00, 1.00) | 1.00 (0.95, 1.01) | 0.8[2] |
| **HKSJ/DL ratio**, Median (IQR) | 1.68 (1.22, 2.90) | 1.19 (0.64, 2.07) | 2.05 (1.65, 6.47) | <0.001[2] |
| **HKSJ and DL confidence interval length comparison** | Smaller | 30(42.25%) | 0 | <0.001[1] |
| | Larger | 41 (57.75%) | 75(100%) | |
| Effect size **DL, Median (IQR)** | 0.14 (-0.14, 1.12) | 0.14 (-0.01, 0.69) | 0.25 (-0.37, 1.31) | |
| Effect size REML, Median (IQR) | 0.14 (-0.14, 1.11) | 0.14 (-0.01, 0.69) | 0.25 (-0.37, 1.22) | |
| Effect size PM, Median (IQR) | 0.14 (-0.14, 1.09) | 0.14 (-0.01, 0.69) | 0.24 (-0.37, 1.22) | |
| Effect sizeHKSJ, Median (IQR) | 0.14 (-0.14, 1.12) | 0.14 (-0.01, 0.69) | 0.25 (-0.37, 1.31) | |
| **P value difference RMEL/DL (mean ± sd) (95%CI)** | 0.00004 ± 0.0558 (95%CI: -0.11, 0.11) | 0.99[2] | 0.0013 ± 0.041 (-0.08,0.08) | 0.97[2] |
| **P value difference PM/DL (mean ± sd) (95%CI)** | 0.00004 ± 0.0558 (-0.11, 0.11) | 0.99[2] | 0.0023 ±0.041(-0.08, 0.08) | 0.95[2] |
| **P value difference HKSJ/DL (mean ± sd) (95%CI)** | -0.001±.052(-0.001,0.2) | 0.051[2] | -0.097± 0.04 (-017, -0.016) | 0.01[2] |

[1] Fisher's exact test

[2] Wilcoxon rank sum test

were conducted using RevMan or CMA, where the default method of between-study variance is DL. Previous simulation studies [26–28] reported that the DL method has a high rate of false positive results. Likewise, two methodological studies [13, 29] found a considerable number of false positive results in Cochrane MAs with significant findings using the DL method. The current investigation replicated 146 MAs with few primary studies less than 5 using four between-study variance methods (DL, REML, PM, HKSJ) and found that (30/69; 43.47%) of the significant MAs became non-significant after applying HKSJ. This finding confirms previous empirical assessments [13, 29], which investigated Cochrane MAs, and revealed that greater significant results resulted from methods based on the normal distribution (such as DL) than methods based on t-distribution (such as HKSJ). Likewise, a simulation study found that t-distribution produces a wider confidence interval than the normal distribution, especially when

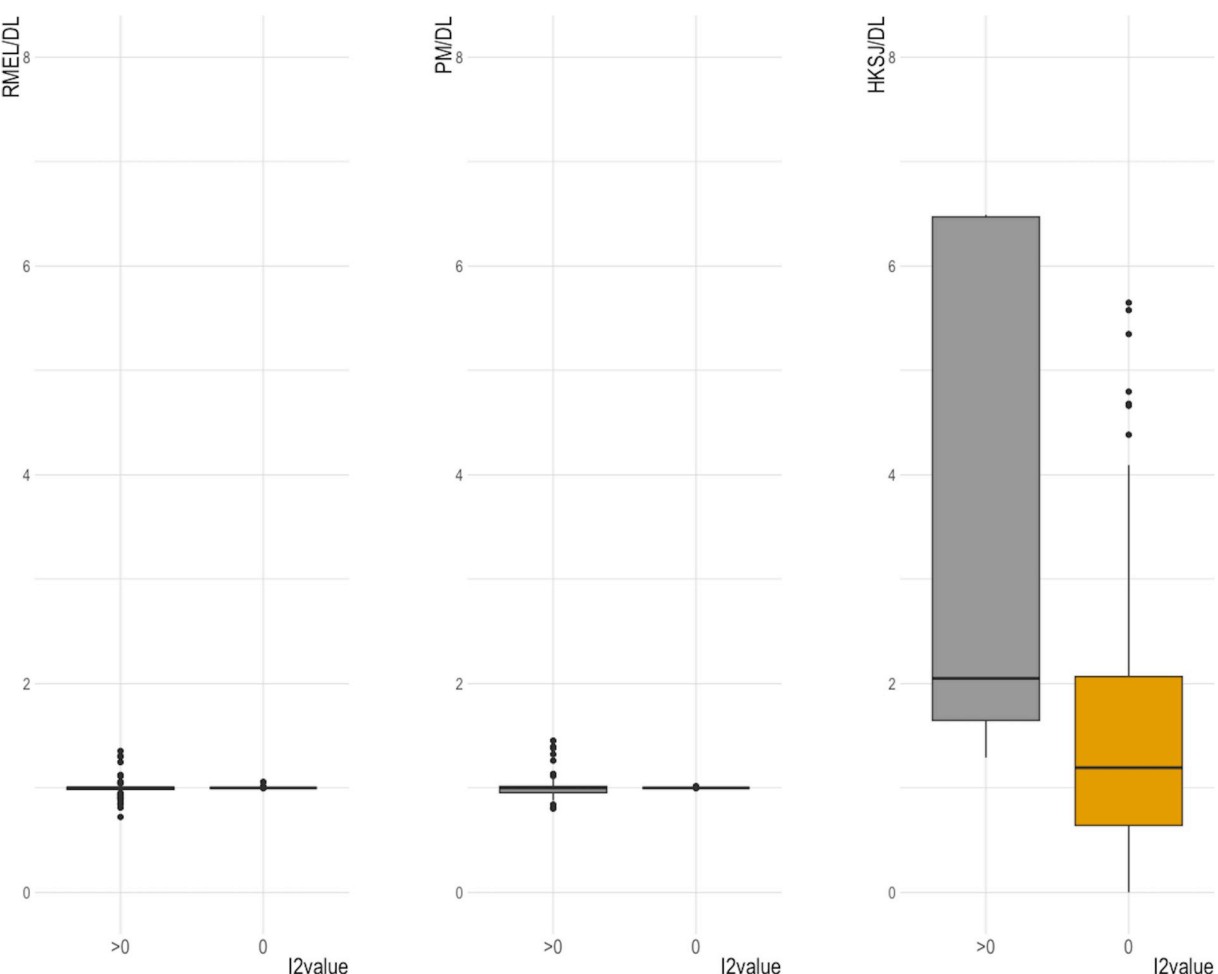

**Fig 2. Boxplot depicts the different ratios of confidence intervals for the three heterogeneity estimators; HKSJ/DL, PM/DL, and REML/DL.** The median of REML/DL and PM/DL is almost equal to one. The median of HKSJ/DL ratio is slightly greater than two when $I^2 = 0\%$, and it is slightly greater than one when $I^2 = 0\%$. It is worth noting that almost one-third of the boxplot represents less than one.

**Table 3. Absolute and relative frequencies of the significance of the meta-analyses using the four different heterogeneity methods.**

| | | Level of Statistical Heterogeneity | | | | | |
|---|---|---|---|---|---|---|---|
| | | $I^2 = 0$ | | | $I^2 > 0$ | | |
| Estimator | DL | Non-significant N = 47 (%) | Significant N = 24 (%) | Total N = 71 (%) | Non-significant N = 40 | Significant N = 35 (%) | Total N = 75 (%) |
| REML | Non-significant | 47 | 0 | 47 (66.20%) | 40 | 0 | 40 (53.33%) |
| | significant | 0 | 24 | 24 (33.80%) | 0 | 35 | 35 (46.67%) |
| PM | Non-significant | 47 | 0 | 47 (66.20%) | 40 | 1 (2.44%) | 41 (54.67%) |
| | significant | 0 | 24 | 24(33.80%) | 0 | 34 (97.14%) | 34 (45.33%) |
| HKSJ | Non-significant | 40(83.33%) | 8(16.67%) | 48 (67.61%) | 40 | 21(60%) | 61 (81.33%) |
| | significant | 7(30.43%) | 16(69.57%) | 23 (32.39%) | 0 | 14(40%) | 14 (18.67%) |

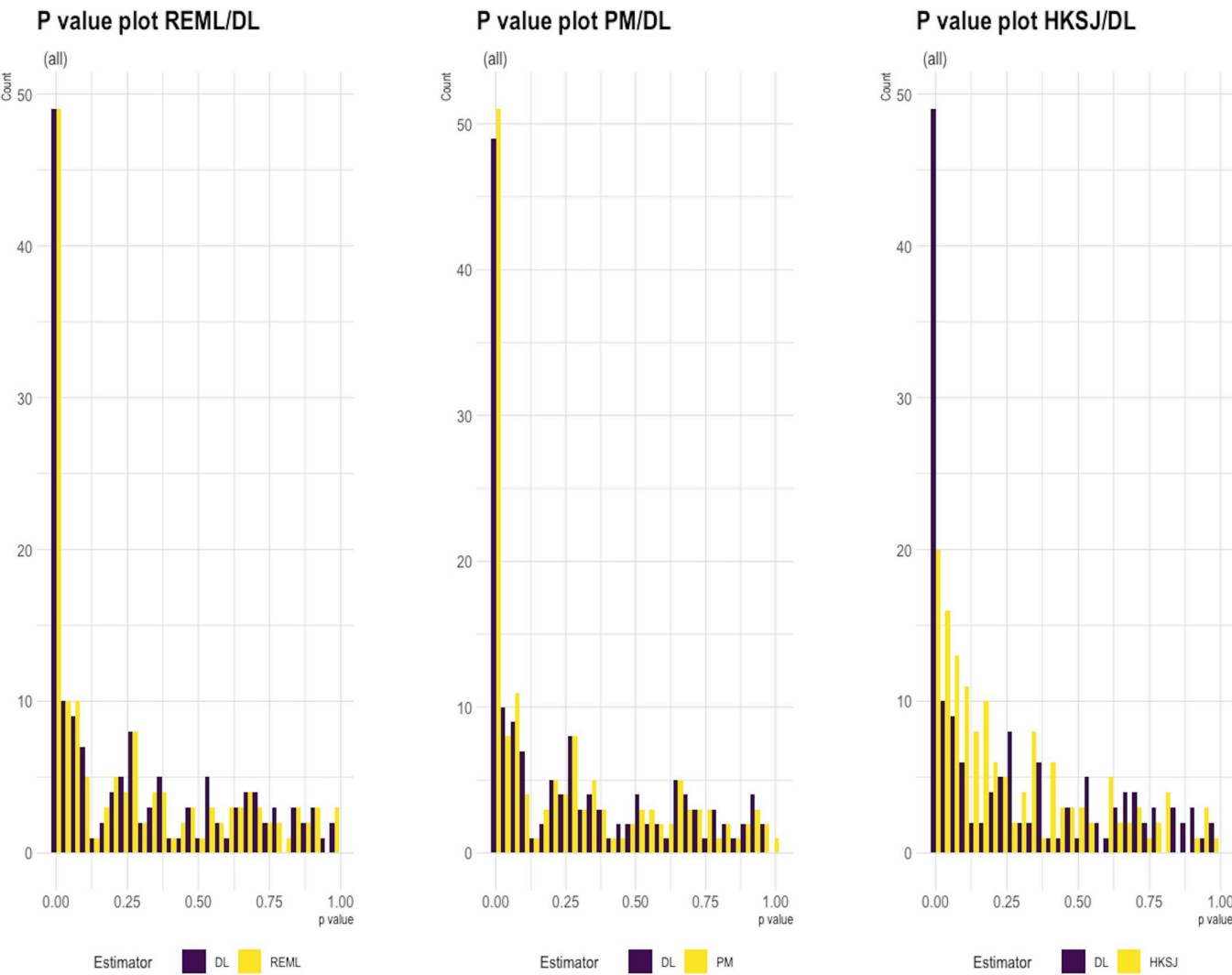

**Fig 3. P-value distribution between the three methods of Heterogeneity compared to DL method.**

the statistical heterogeneity and the number of studies are small in meta-analysis [30]. This may rias concerns related to the overcoverage and loss of power [31]. The aforementioned information my explain our findings, as the confidence interval of MAs using HKSJ was double the length up to approximately 7 times the length of the confidence interval when DL was used, and the heterogeneity was present ($I^2 > 0\%$). Hence, HKSJ is over-conservative and may result in non-significant results when they are actually significant. A strength of this study is comparing the results of the different methods based on the heterogeneity level.

It is also important to mention that 30.43% (7/40) of non-significant MAs using the DL method became significant with HKSJ when the statistical heterogeneity was absent ($I^2 = 0\%$). This can be supported by the finding that the confidence interval was narrower in 42.25% (30/75) of MAs using HKSJ than that using DL when the statistical heterogeneity was absent ($I^2 = 0\%$).(Fig 5) These findings were consistent with a previous study [32] which replicated the analysis of 157 MAs with binary data using HK and DL and concluded that HK may yield a narrower confidence interval and smaller P-values than DL in some homogeneous meta-analyses. Consequently, HKSJ is not always conservative [31] and may also be imperfect and may

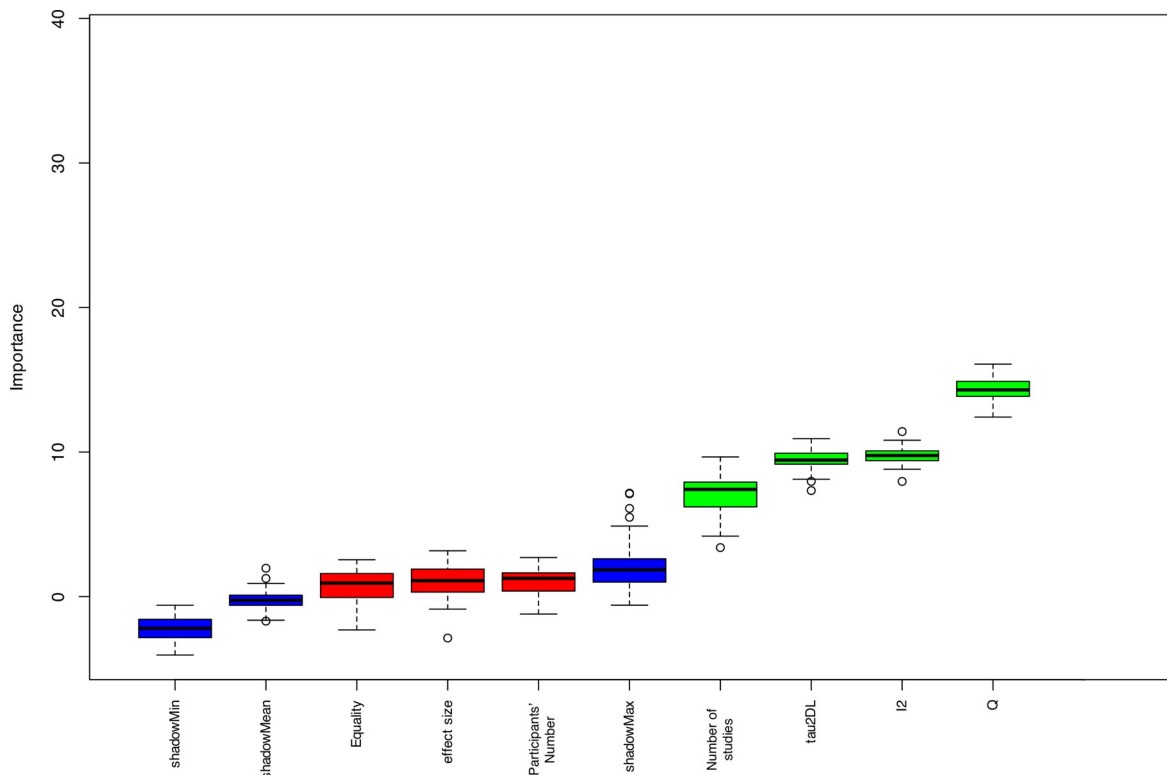

**Fig 4. The Boruta algorithm model shows significant predictors with green boxplots.** Red boxplots represent the insignificant features. The blue boxplots represent the shadow attributes.

result in false positive results when heterogeneity is absent. A previous study [31] suggested to avoid Hartung-Knapp method When the heterogeneity is absent ($t^2 = 0$) due to producing a modified results with shorter confidence intervals and smaller P values.

The Boruta algorithm found that the number of studies and the heterogeneity measures are significant predictors for the change in HKSJ confidence interval width. However, the model rejected the effect of equality of trials' size on the confidence interval change between HKSJ and DL methods. In contrast, a previous study [13] found that equal-sized trials may decrease the change in type I error rate even if there is a moderate heterogeneity. In our study, the high number of unequal-size trials, the large difference in trials' size in the individual MA, or the very few trials in MA might render the effect of the size equality trivial.

Simulation studies [33] showed that REML and PM methods are more robust than the DL method. They recommended the PM method for both continuous and binary outcomes and the REML method for continuous outcomes [34] even for large heterogeneity ($t^2$). A recent study found similar results for DL, REML, and PM, especially when the heterogeneity was absent. This is consistent with the findings of Chung et al. [35], who found that DL and REML produce similar findings when the number of studies is small.

A previous investigation [24] included a wide range of pooled studies in a meta-analysis and assessed the results using eight different heterogeneity methods with and without HK adjustment. The authors concluded that "meta-analysis with at least three studies is sensitive to HK correction". However, we found that in meta-analysis with few studies (fewer than 5), HKSJ is overconservative when $I^2$ is greater than 0%, while HKSJ may result in false positive results when the heterogeneity is absent ($I^2 = 0\%$). Although making a clear recommendation

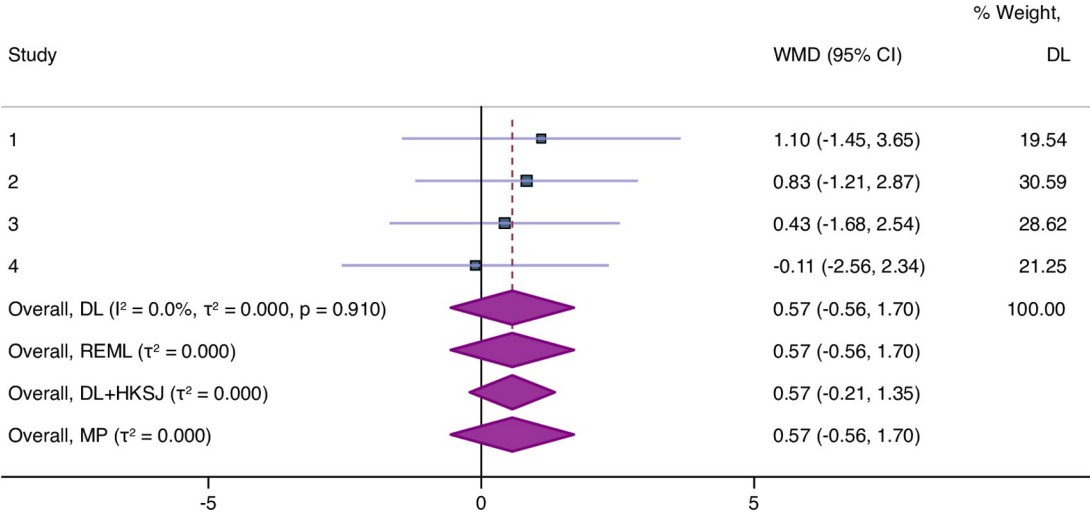

**Fig 5. Example of replicated meta-analysis for four trials with a lack of statistical heterogeneity ($I^2 = 0$%).** The confidence interval is equal in the three methods (DL, REML, and PM) (95%CI; -0.56 to 1.70), but it is narrower when HKSJ estimator was applied (95%CI; -0.21 to 1.35).

about the best methods is difficult in case of few studies, we suggest using DL method for pooling a small number of primary studies (fewer than 5) in MAs when $I^2 = 0$% and sensitivity analysis using both HKSJ and DL for pooling MAs when $I^2 > 0$.

## Clinical implications

The clinical implications of this study are important for MAs focused on orthodontic samples. As many MAs related to orthodontics include small samples of primary studies, selecting the between-study variance methods when conducting the MAs can determine whether the identified differences are significant. It is recommended that MAs related to orthodontics that include small samples of primary studies should report using more than one between-study variance method (sensitivity analysis), especially when the confidence interval is close to the significance/non-significance cut-off point.

## Limitations

The search period was restricted to three years, but the study aimed to collect MAs rather than only SRs and to map a problem rather than providing a more robust estimate. Although relatively short, this period probably expresses a reality closer to what has been investigated nowadays. MAs with only two to four studies were included in this study to concentrate on MAs with fewer primary studies. Finally, our assessment did not investigate the effect of between-study variance methods on the prediction intervals because of approximately more than one-third of the sample (53/146; 36.3%) had only two studies.

## Conclusion

This sample of MAs with less than five studies appears to be sensitive to the selected between-study variance method. HKSJ doubled the confidence interval of the pooled estimate when the heterogeneity was present. However, HKSJ reduced the confidence intervals of 30% of MAs when the heterogeneity was absent, leading to more significant differences when compared to the DL method.

## Supporting information

**S1 Table. Search strategy in PubMed.**
(DOCX)

**S2 Table. Excluded studies with reason.**
(DOCX)

**S1 Checklist. PRISMA 2020 checklist.**
(DOCX)

## Author Contributions

**Conceptualization:** Samer Mheissen, Haris Khan, David Normando, Nikhillesh Vaiid, Carlos Flores-Mir.

**Data curation:** Samer Mheissen, Haris Khan.

**Formal analysis:** Samer Mheissen.

**Investigation:** Haris Khan, David Normando, Nikhillesh Vaiid.

**Methodology:** Samer Mheissen, Haris Khan, Nikhillesh Vaiid, Carlos Flores-Mir.

**Project administration:** David Normando, Carlos Flores-Mir.

**Resources:** David Normando, Nikhillesh Vaiid.

**Software:** Samer Mheissen, David Normando.

**Supervision:** Carlos Flores-Mir.

**Validation:** Nikhillesh Vaiid.

**Visualization:** David Normando, Nikhillesh Vaiid.

**Writing – original draft:** Samer Mheissen, Haris Khan, Carlos Flores-Mir.

**Writing – review & editing:** Samer Mheissen, David Normando, Nikhillesh Vaiid, Carlos Flores-Mir.

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
