## [Decision Letter · Decision Letter 0]

15 Jan 2024

PONE-D-23-41795

Do Statistical Heterogeneity methods impact the results of Meta- Analyses? A Meta Epidemiological Study.

PLOS ONE

Dear Dr. Mheissen,

Thank you for submitting your manuscript to PLOS ONE. After careful consideration, we feel that it has merit but does not fully meet PLOS ONE’s publication criteria as it currently stands. Therefore, we invite you to submit a revised version of the manuscript that addresses the points raised during the review process.

We look forward to receiving your revised manuscript.

Kind regards,

Ashraful Hoque, MD

Academic Editor

PLOS ONE

Journal Requirements:

3. We note that this manuscript is a systematic review or meta-analysis; our author guidelines therefore require that you use PRISMA guidance to help improve reporting quality of this type of study. Please upload copies of the completed PRISMA checklist as Supporting Information with a file name “PRISMA checklist”.

Reviewers' comments:

Reviewer's Responses to Questions

**Comments to the Author**

1. Is the manuscript technically sound, and do the data support the conclusions?

Reviewer #1: Yes

Reviewer #2: Yes

2. Has the statistical analysis been performed appropriately and rigorously? 

Reviewer #1: I Don't Know

Reviewer #2: Yes

3. Have the authors made all data underlying the findings in their manuscript fully available?

Reviewer #1: Yes

Reviewer #2: Yes

4. Is the manuscript presented in an intelligible fashion and written in standard English?

Reviewer #1: Yes

Reviewer #2: Yes

5. Review Comments to the Author

Reviewer #1: Abstract: In the results section of your abstract you wrote (n=111, 76%). This is unclear. Use a semicolon instead of a comma.

I notice here that you give different decimal places. The manuscript should be uniform. The same decimal places should be given throughout the manuscript.

Is the “Clinical relevance” heading in the abstract necessary? If recommend deleting it since you have stated your conclusion already.

Main text:

What I am missing is a meta-analyses quality assessment, using AMSTAR 2 (A MeaSurement Tool to Assess Systematic Reviews 2).

Reviewer #2: The authors aimed to summarise the impact of different estimators on the results of meta-analyses with less than five studies in the field of orthodontics. Below I provide some comments in which I believe can improve the manuscript.

[1] Abstract | Please cross-check the abstract to PRISMA abstract guidelines. EG dates of searches should be mentioned.

[2] Methods | Given this was a systematic review of systematic reviews I would suggest formatting the paper and report in line with PRISMA guidelines. See here: http://www.prisma-statement.org/

[3] Results – Characteristics | Regarding, “Likewise, most of the included MAs did not report the

heterogeneity estimator (n=99, 83%)”, which heterogeneity estimator are you referring to, or are you referring to all estimators? Please clarify.

[4] Results – Characteristics | RE “The most frequently used software was RevMan (n=112; 77%),”. My understanding if that RevMan automatically uses the DL methods of estimation. Above, you note that DL was only report in n=14 reviews. Could it be assumed that most studies used the DL method if this is the case? I think this may be worth clarifying if so here.

[5] Results/Discussion | You mention that the confidence intervals were narrower when heterogeneity was zero for HJSK. Sometimes ad-hoc variance correction is recommended in these cases (e.g., using the wider of the confidence intervals): https://pubmed.ncbi.nlm.nih.gov/28748567/. I think this should be highlighted further in the discussion in the second paragraph.

6. PLOS authors have the option to publish the peer review history of their article (what does this mean?). If published, this will include your full peer review and any attached files.

Reviewer #1: No

Reviewer #2: No

---

## [Author Response · Author response to Decision Letter 0]

18 Jan 2024

Dear Editor Ashraful Hoque,

We thank you and the reviewers for the comments and suggestions that will improve our manuscript. Please find below the responses and actions taken.

Reviewers' Comments to Author:

Reviewer #1: Abstract: In the results section of your abstract, you wrote (n=111, 76%). This is unclear. Use a semicolon instead of a comma.

Author response: We have amended the abstract accordingly.

I notice here that you give different decimal places. The manuscript should be uniform. The same decimal places should be given throughout the manuscript.

Author response: All decimal places have been amended to two decimal places in the tables and throughout the manuscript except for the p-value which is less than 1 and the other decimals are important.

Is the “Clinical relevance” heading in the abstract necessary? If recommend deleting it since you have stated your conclusion already.

Author response: the clinical relevance heading was removed according to the reviewer’s comment.

Main text:

What I am missing is a meta-analyses quality assessment, using AMSTAR 2 (A MeaSurement Tool to Assess Systematic Reviews 2).

Author response: We appreciate the reviewer's input. However, a recent methodological study has already evaluated the methodological quality of orthodontic systematic reviews(1) in 2021. Consequently, we opted not to include this assessment to avoid duplicating research efforts. Additionally, it is important to note that the primary aim of our study was to examine the impact of different tau2 estimators on the statistical significance of results and confidence intervals rather than focusing on the methodological quality of the included studies.

Reviewer #2: The authors aimed to summarise the impact of different estimators on the results of meta-analyses with less than five studies in the field of orthodontics. Below I provide some comments in which I believe can improve the manuscript.

We want to thank you and the reviewers for the comments and suggestions that we think will improve our manuscript. 

[1] Abstract | Please cross-check the abstract to PRISMA abstract guidelines. EG dates of searches should be mentioned.

Author response: the search date was added to the abstract, and the structure of the abstract was changed to follow PRISMA abstract guidelines.

[2] Methods | Given this was a systematic review of systematic reviews I would suggest formatting the paper and report in line with PRISMA guidelines. See here: http://www.prisma-statement.org/

Author response: We attempted to report the paper according to modified PRISMA guidelines for methodological studies.(2)

[3] Results – Characteristics | Regarding, “Likewise, most of the included MAs did not report the

heterogeneity estimator (n=99, 83%)”, which heterogeneity estimator are you referring to, or are you referring to all estimators? Please clarify.

Author response: we are referring to the heterogeneity estimator, whatever it was. The line was amended accordingly. 

[4] Results – Characteristics | RE “The most frequently used software was RevMan (n=112; 77%),”. My understanding if that RevMan automatically uses the DL methods of estimation. Above, you note that DL was only report in n=14 reviews. Could it be assumed that most studies used the DL method if this is the case? I think this may be worth clarifying if so here.

Author response: the reporting of the heterogeneity method reflects the authors' knowledge. Hence, we tried to examine the reporting method used. We concur with the reviewer's observation that DL was the default method in most of our samples. We clarified this in the first paragraph of the discussion: Although there was a lack of reporting of the between-study variance method in 83% (99/119) of random meta-analyses, 98% (97/99) of those MAs were conducted using RevMan or CMA, where the default method of between-study variance is DL.

[5] Results/Discussion | You mention that the confidence intervals were narrower when heterogeneity was zero for HJSK. Sometimes ad-hoc variance correction is recommended in these cases (e.g., using the wider of the confidence intervals): https://pubmed.ncbi.nlm.nih.gov/28748567/. I think this should be highlighted further in the discussion in the second paragraph.

Author response: Thank you for bringing this to our attention. We have accordingly revised the discussion. Additionally, the last paragraph in the discussion aligns with the findings of the mentioned study, advocating for the use the modified methods as sensitivity analysis:

Although making a clear recommendation about the best methods is difficult in case of few studies, we suggest using DL method for pooling a small number of primary studies (fewer than 5) in MAs when I2=0% and sensitivity analysis using both HKSJ and DL for pooling MAs when I2>0.

Reference 

1. Hooper EJ, Pandis N, Cobourne MT, Seehra J. (2021) Methodological quality and risk of bias in orthodontic systematic reviews using AMSTAR and ROBIS. Eur J Orthod, 43: 544-550

2. Murad MH, Wang Z. (2017) Guidelines for reporting meta-epidemiological methodology research. Evid Based Med, 22: 139-142

---

## [Editor Report · Acceptance letter]

3 Mar 2024

PONE-D-23-41795R1 

PLOS ONE

Dear Dr. Mheissen, 

I'm pleased to inform you that your manuscript has been deemed suitable for publication in PLOS ONE. Congratulations! Your manuscript is now being handed over to our production team.

Kind regards, 

on behalf of

Dr. Ashraful Hoque 

Academic Editor

PLOS ONE